# Rapamycin administration causes a decrease in muscle contractile function and systemic glucose intolerance concomitant with reduced skeletal muscle Rictor, the mTORC2 component, expression independent of energy intake in young rats

Satoru Ato[1,2,3]*, Chieri Oya[1], Riki Ogasawara[1,2†]

1 Department of Life Science and Applied Chemistry, Nagoya Institute of Technology, Nagoya, Japan,
2 Healthy Food Science Research Group, Cellular and Molecular Biotechnology Research Institute, National Institute of Advanced Industrial Science and Technology (AIST), Tsukuba, Japan, 3 Faculty of Health and Sports Sciences, Toyo University, Tokyo, Japan

† Deceased.
* satorugby@gmail.com

**Data Availability Statement:** All relevant data are within the manuscript and its Supporting

## Abstract

Emerging evidence suggests the potential of rapamycin, an antibiotic from *Streptomyces hygroscopicus* that functions as a mechanistic target of rapamycin (mTOR) inhibitor, as a mimetic of caloric restriction (CR) for maintaining skeletal muscle health. Several studies showed that rapamycin administration (RAP) reduced appetite and energy intake. However, the physiological and molecular differences between RAP and CR in skeletal muscle are not fully understood. Here we observed the effects of 4 weeks of RAP administration and CR corresponding to the reduction in energy intake produced by RAP administration (PF, paired feeding) on fast glycolytic and slow oxidative muscle in young adult rats. We found that 4 weeks of RAP demonstrated low fast-glycolytic muscle mass with smaller type I and IIb/x myofiber size independent of the energy intake. In addition, PF improved the contractile function of the plantar flexor muscle, whereas RAP did not improve its function. The suppressing response of mTORC1 signaling to RAP is greater in slow-oxidative muscles than in fast-glycolytic muscles. In addition, systemic glucose tolerance was exacerbated by RAP, with reduced expression of Rictor and hexokinase in skeletal muscle. These observations imply that RAP may have a slight but significant negative impact and it obviously different to CR in young adult skeletal muscle.

## Introduction

Skeletal muscle, which is widely distributed throughout the body, functions as a locomotor organ and is the largest energy-consuming organ for glucose utilization. Emerging evidence

Information files. https://doi.org/10.6084/m9.
figshare.27086701.v2

**Funding:** The author(s) received no specific funding for this work.

**Competing interests:** The authors have declared that no competing interests exist.

indicates that low skeletal muscle mass is an independent risk factor for death [1, 2]. Skeletal muscle loss occurs under several conditions, including aging, cancer cachexia, and type 2 diabetes [3]. It is important to understand the mechanisms regulating skeletal muscle mass to develop an efficient strategy for maintaining or improving skeletal muscle mass.

Skeletal muscle mass is considered to be maintained by regulating the balance between the synthesis and breakdown of muscle protein, and energy intake is essential for maintaining muscle protein synthesis (MPS) [4]. Surprisingly, a recent study has shown that calorie restriction has numerous beneficial effects on skeletal muscle function, such as improved mitochondrial quality [5–8] and satellite cell function [9] and reduced aging-associated loss of muscle mass [10, 11].

Interestingly, administration of rapamycin, an antibiotic from *Streptomyces hygroscopicus*, has been reported to elicit effects that are highly analogous to those observed under calorie restriction, especially against aging [12, 13]. The serine/threonine kinase mechanistic target of rapamycin (mTOR) functions as a positive regulator of protein translation through at least two distinct molecular complexes, mTOR complex1 (mTORC1) and mTORC2 [14]. mTORC1 is activated with nutritional stimulus (i.e., by amino acids and insulin), and rapamycin suppresses nutritional activation of mTORC1 in skeletal muscle [15, 16]. Administration of rapamycin has also been reported to reduce appetite and concomitant energy intake [17, 18], suggesting that the effects of rapamycin on skeletal muscle may partially result from caloric restriction due to reduced dietary intake. Chronic rapamycin administration has been reported to cause whole-body insulin resistance by reducing mTORC2 in several insulin-sensitive tissues (e.g., liver, adipose tissue, and skeletal muscle) [12], suggesting that the effects of rapamycin administration and energy restriction on skeletal muscle are different, and rapamycin administration might negatively impact skeletal muscle and whole-body glucose metabolism. Furthermore, a recent functional analysis indicated that both mTORC1 and mTORC2 are important for maintaining proper muscle function and metabolism [19–21]. Although the effect of rapamycin administration has been evaluated in aging conditions, a few studies imply that rapamycin treatment negatively impacts the skeletal muscle in young and non-pathological conditions [17]. However, the chronic effects of rapamycin on skeletal muscle in young rats remain unclear. Understanding this would be informative for effective pharmacological or dietary treatment to improve skeletal muscle. Here, we investigated the effects of chronic rapamycin administration on skeletal muscle considering the reduction in energy intake accompanied by rapamycin administration.

## Results

### Body composition and blood biochemistry

The Rapamycin administration (RAP) group showed blunted body weight growth over time with ~22% reduced spontaneous feeding (Fig 1A and 1B). Pair-feed (PF) group also exhibited similar changes in the body weight, suggesting that body weight reduction due to rapamycin administration is dependent on reduced feeding behavior. Epididymal fat mass was significantly lower in the RAP and PF groups than that in the control (CON) group (P <0.05, Fig 1C). Fasting serum insulin concentration was significantly lower in the RAP group as compared with the CON and PF group (P <0.05, Fig 1D).

In addition, IPGTT demonstrated that the RAP group has glucose intolerance, despite the fasting basal blood glucose level being similar between the groups (Fig 1E and 1F).

### Skeletal muscle characteristics

Absolute gastrocnemius muscle mass was significantly different between groups (Fig 2A). RAP group showed significantly lower values as compared with CON and PF group (-13.5% and

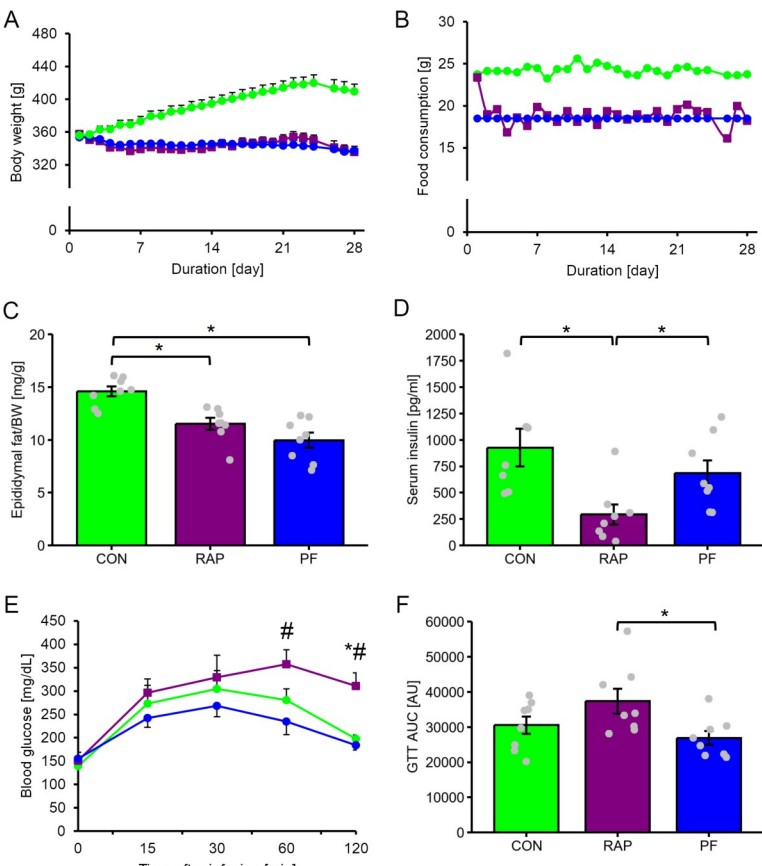

**Fig 1. The influence of 4-week of rapamycin administration on body composition and blood biochemistry.** A. Changes in body weight during experimental period, white circle: CON group, black square: RAP group, gray circle: PF group. B. Changes in food consumption during experimental period, white circle: CON group, black square: RAP group. C. Epididymal fat weight at the post intervention period. D. Serum insulin concentration at the post intervention period. E. Changes in blood glucose level after intraperitoneal glucose tolerance test (IPGTT). F. Area under the curve (AUC) of the changes in blood glucose level after intraperitoneal glucose tolerance test (IPGTT). n = 8 in each group. Values are expressed as MEANS ± SE with individual value (only bar chart). *: P <0.05 vs. CON group. #: P <0.05 vs. PF group. Bars with asterisks indicate significant differences (P <0.05) between two linked factors.

-7.2% respectively, P <0.05). PF group showed significantly lower than CON group (-6.9%, P <0.05). On the contrary, gastrocnemius muscle mass normalized by body weight in the RAP group was significantly higher than in the CON group (P <0.05, Fig 2B). In addition, the PF group showed significantly higher normalized gastrocnemius muscle mass than the CON and RAP groups (P <0.05). Absolute soleus muscle mass did not differ between groups (Fig 2C). On the contrary, body weight-normalized soleus muscle in the RAP group showed a significantly higher value than in the CON group (P <0.05, Fig 2D).

Myofiber CSA for each fiber type was measured from the gastrocnemius muscle. RAP group showed significantly smaller type 1 fiber CSA than the PF group (Fig 2E and 2F). Type 2x/b and mean fiber CSA (all types of fiber CSA averaged) was also smaller in RAP group as compared to other two groups. In addition, the PF group showed smaller Type 2x/b and mean fiber CSA than that of the CON group (Fig 2F).

The composition of MHC type was not significantly different between the group and both glycolytic-gastrocnemius muscle and oxidative-soleus muscle (Fig 2G).

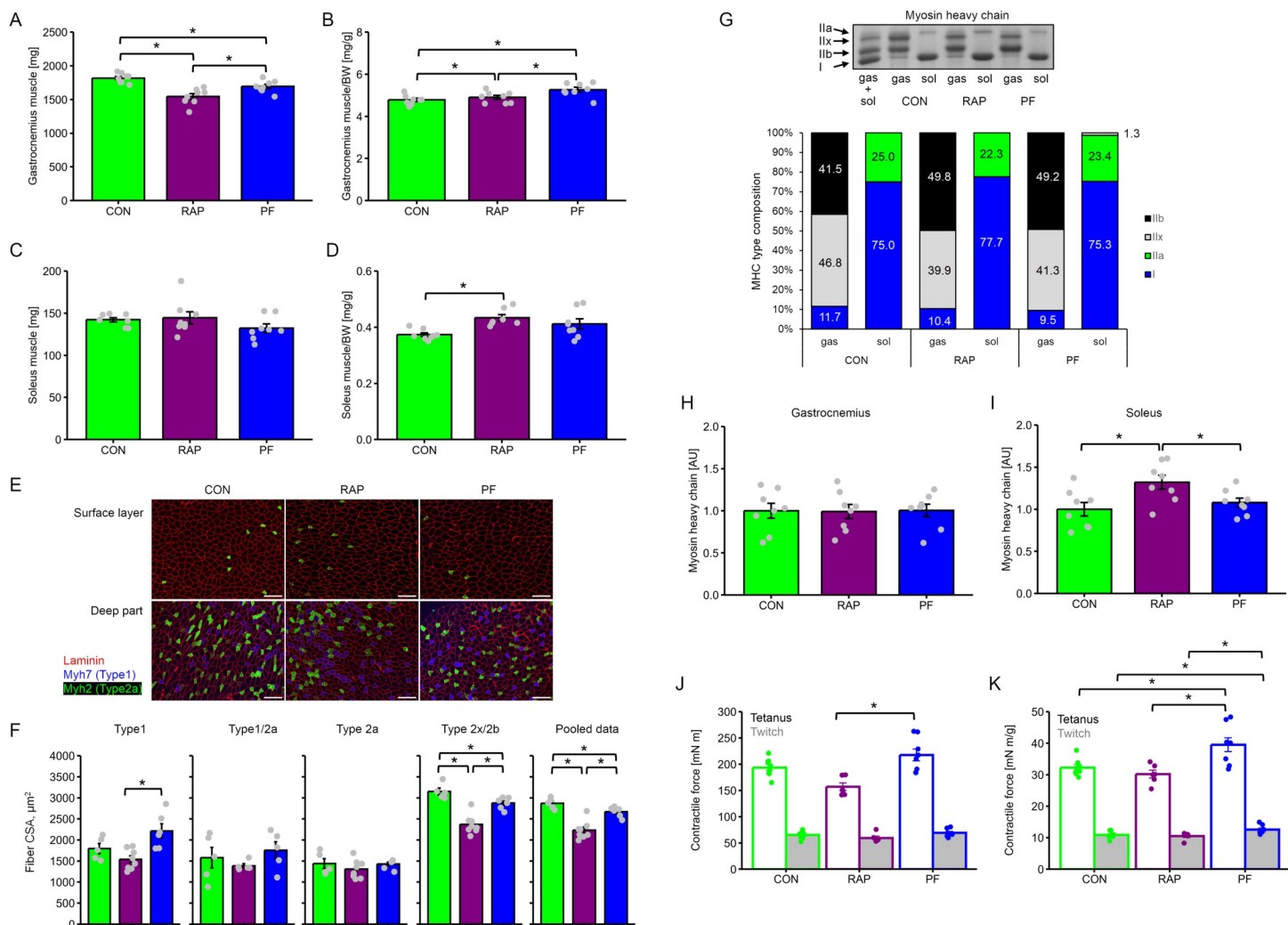

**Fig 2. The influence of 4-week of rapamycin administration on skeletal muscle characteristics.** A. gastrocnemius muscle weight at the post intervention period in CON, RAP, and PF group. B. Body weight normalized gastrocnemius muscle weight (MW) at the post intervention period in CON, RAP, and PF group. C. Soleus muscle weight at the post intervention period in CON, RAP, and PF group. D. Body weight normalized soleus muscle weight at the post intervention period in CON, RAP, and PF group. E. Representative cross-section image of the gastrocnemius muscle labeled with Laminin, Myh7 (type1), and Myh2 (type2a) antibodies. Scale bar: 100 μm. F. Myofiber CSA for each fiber type. G. Myosin heavy chain (MHC) type composition calculated from MHC typing with SDS-PAGE (upper: representative gel image). The number inside the bars indicates the percentage of each MHC type. H. Myosin protein expression in gastrocnemius muscle calculated from gel electrophoresis. I. Myosin protein expression in soleus muscle calculated from gel electrophoresis. J. plantar flexor force. K. plantar flexor force normalized with triceps surae muscle weight. n = 8 in each group. Values are expressed as MEANS ± SE with individual value (only bar chart). Bars with asterisks indicate significant differences (P <0.05) between two linked factors.

MCH content in gastrocnemius muscle was not significantly different between groups ([Fig 2H]), while MHC content in soleus was significantly higher in the RAP group as compared with CON and PF group ([Fig 2I], P <0.05 respectively).

Absolute tetanic force ([Fig 2J]) was significantly lower in the RAP group as compared with PF (P <0.05), while the value in twitch force did not differ between the group. Normalized-tetanus force by triceps surae muscle weight ([Fig 2K]) was significantly higher in the PF group as compared with CON and RAP groups, respectively (P <0.05). Similarly, Normalized-twitch force was significantly higher in the PF group as compared with CON and RAP groups, respectively (P <0.05).

## Component of mTOR complex and substrates of mTORC1 signaling

mTOR expression was not different between group and muscle type (Fig 3A). Level of FKBP12 was not significantly different between the group, but significantly more abundant in soleus muscle as compared with gastrocnemius muscle (Fig 3B). Raptor, a mTORC1 component, expression in gastrocnemius muscle was not different between three groups, while raptor in soleus muscle was significantly higher in PF group as compared with CON group (Fig 3C, P <0.05). While Rictor, a mTORC2 component, expression level did not differ between muscle types (Fig 3D), the RAP group demonstrated significantly lower Rictor expression than in both types of muscle in the CON and PF group (P <0.05).

The phosphorylation level of p70S6K Thr389 (Fig 3E), a direct phosphorylation residue of mTORC1, is higher in soleus muscle as compared with gastrocnemius muscle (P <0.05). RAP

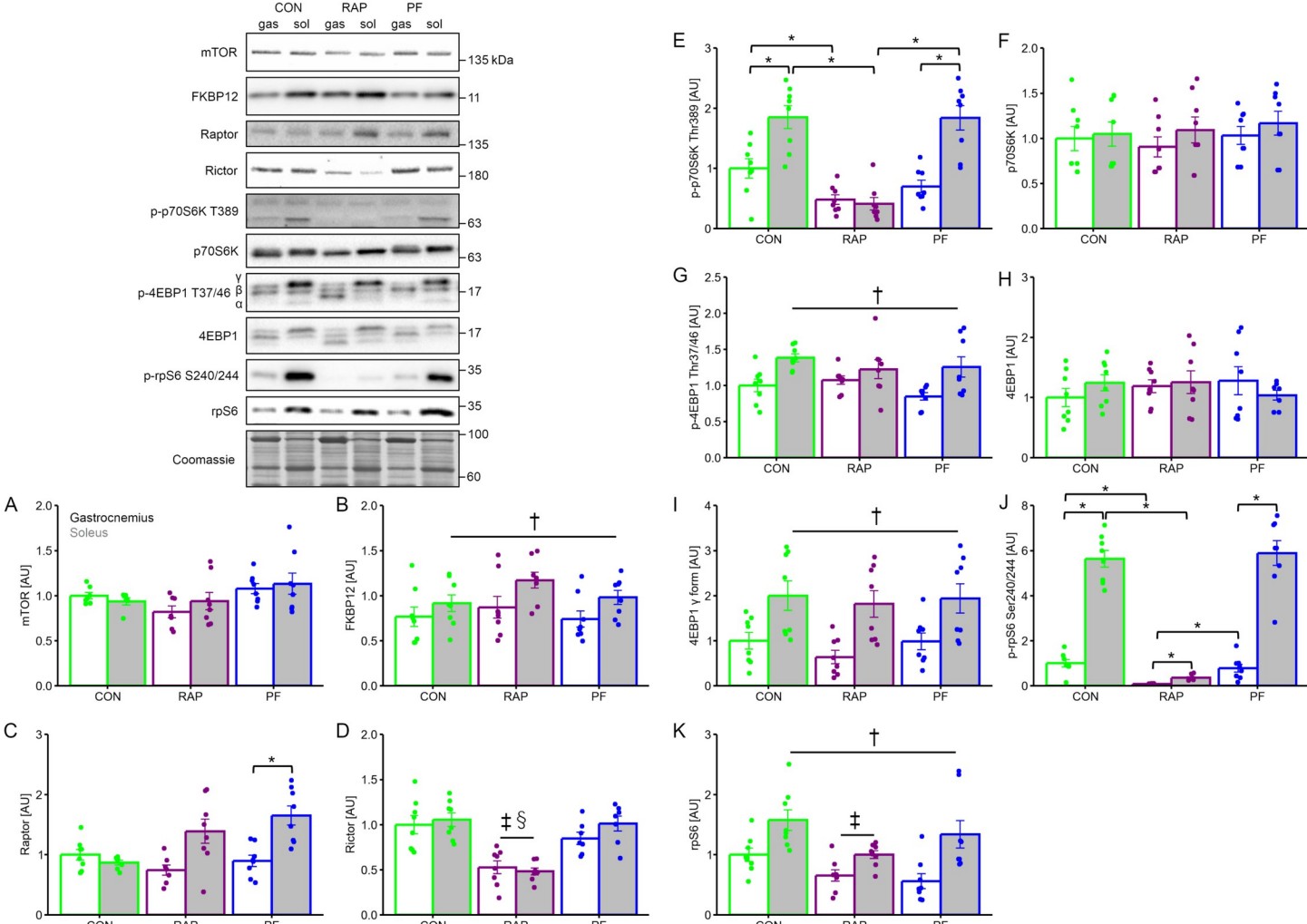

**Fig 3. The influence of 4-week of rapamycin administration on the mTOR signaling in gastrocnemius and soleus muscle.** A. mTOR protein expression. B. FKBP12 protein expression. C. Raptor protein expression. D. Rictor protein expression. E. Thr389 phosphorylated p70S6K protein expression. F. p70S6K protein expression. G. Thr37/46 phosphorylated 4EBP1 protein expression. H. 4EBP1 protein expression. I. γ form (hyperphosphorylated) of 4EBP1 protein expression. J. Ser240/244 phosphorylated rpS6 protein expression. K. rpS6 protein expression. n = 8 in each group. Values are expressed as MEANS ± SE with individual value. ME: main effect. †: P <0.05, gastrocnemius vs. soleus (main effect of muscle type by 2-way ANOVA), ‡: P <0.05 vs. CON group (main effect of group by 2-way ANOVA, comparison between groups). §: P <0.05 vs. PF (comparison between groups). Different character indicates statistically significant difference (P <0.05).

group showed significantly lower phosphorylation levels as compared with CON and PF group in both muscle types. p70S6K expression was not significantly different neither the group nor muscle type (Fig 3F).

The phosphorylation level of 4EBP1 Thr37/46 (Fig 3G), another direct phosphorylation residue of mTORC1, is higher in soleus muscle as compared with gastrocnemius muscle (main effect of muscle type, P <0.05). On the contrary, the difference between groups was not observed on the level of 4EBP1 phosphorylation. 4EBP1 expression was not significantly different neither the group nor muscle type (Fig 3H). Gamma (inactivated) form of 4EBP1 expression was significantly greater in soleus muscle than in gastrocnemius muscle (Fig 3I. main effect of muscle type, P <0.05), while it did not differ between the group.

Phosphorylation of rpS6 Ser240/244, a direct phosphorylation residue of p70S6K, was significantly higher in soleus muscle than in gastrocnemius muscle (Fig 3J, P <0.05). RAP group showed significantly lower phosphorylation levels as compared with CON and PF groups in both muscle types (P <0.05). rpS6 expression was significantly greater in soleus muscle as compared with gastrocnemius muscle (Fig 3K. main effect of muscle type, P <0.05). RAP group showed significantly lower rpS6 expression as compared with the CON group (P <0.05).

## Muscle protein synthesis and molecules involved in protein degradation

Basal muscle protein synthesis level was significantly higher in soleus muscle as compared with gastrocnemius muscle (Fig 4. main effect of muscle type, P <0.05), while a significant difference was not seen between the groups.

## Protein degradation pathways

LC3-I, a molecular marker of an initial stage of autophagy, was significantly highly expressed in soleus muscle than in gastrocnemius muscle (Fig 5A. main effect of muscle type, P <0.05). RAP group showed significantly greater LC3-I expression exclusively in soleus muscle as compared with CON and PF group (P <0.05 in each). LC3-II, a molecular marker of an isolation membrane formation, is also significantly highly expressed in soleus muscle than in gastrocnemius muscle (Fig 5B. main effect of muscle type, P <0.05). In addition, the PF group showed a significantly lower LC3-II expression level as compared with the CON and RAP groups (P <0.05 in each). LC3-II / I ratio is significantly higher in soleus muscle as compared with gastrocnemius muscle (Fig 5C. main effect of muscle type, P <0.05). The soleus muscle in the PF group demonstrated a significantly lower LC3-II / I ratio as compared with the soleus muscle in CON and RAP groups (P <0.05 in each). The p62 expression was significantly greater in soleus muscle as compared with gastrocnemius, while it did not differ between groups (Fig 5D).

Ubiquitinated protein expression was significantly greater in soleus muscle as compared with gastrocnemius muscle, while group difference was not observed (Fig 5E). On the contrary, ubiquitinated protein more than 180 kDa in soleus muscle was significantly abundantly observed in the RAP group as compared with PF group and a similar trend (P = 0.1) was observed between CON and RAP group (Fig 5F. main effect of muscle type, P <0.05).

## Glucose metabolism regulators

The level of Akt Thr308 phosphorylation was significantly different between soleus and gastrocnemius muscle in the RAP group (Fig 6A, P <0.05). Akt Thr308 phosphorylation in the soleus muscle was significantly higher in the RAP group as compared with the PF group (Fig 6A, main effect of muscle type P <0.05). In addition, soleus muscle Akt Thr308 phosphorylation in the RAP group was significantly higher than in the PF group (P <0.05). Akt Ser473 phosphorylation was significantly lower in the RAP group than in CON and PF groups (Fig

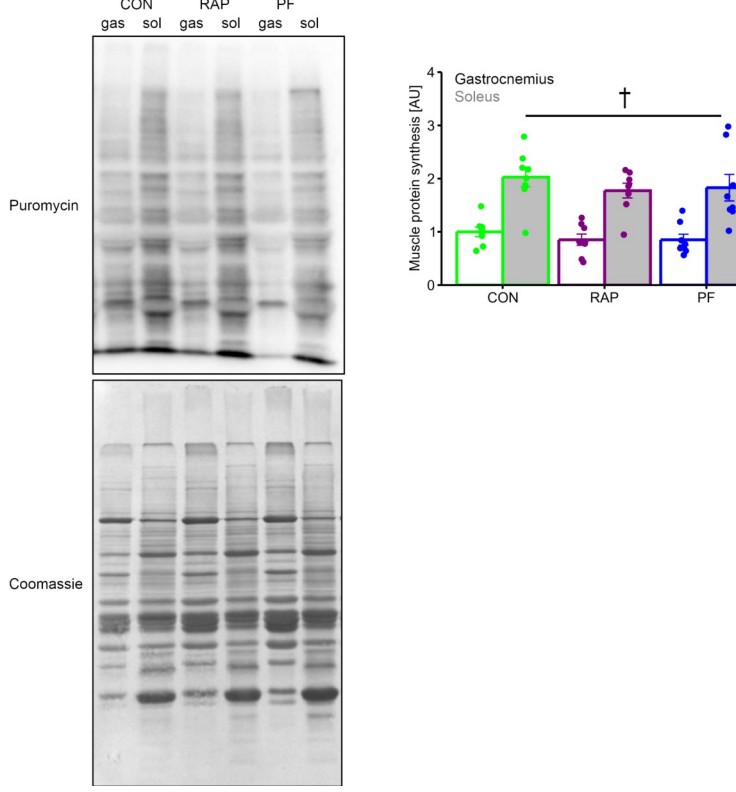

**Fig 4. The influence of 4-week of rapamycin administration on the protein synthesis in gastrocnemius and soleus muscle.** n = 8 in each group. Values are expressed as MEANS ± SE with individual value. †: P <0.05 between gastrocnemius vs. soleus (main effect of muscle type by 2-way ANOVA).

6B, P <0.05 in each). In addition, the level of Akt Ser473 phosphorylation was different between soleus and gastrocnemius muscle in the RAP and PF groups (P <0.05 in each). Akt expression level was significantly greater in soleus muscle as compared with gastrocnemius muscle (Fig 6C, P <0.05). Furthermore, Akt expression in the soleus muscle was significantly lowered in the RAP group as compared with CON and PF group (P <0.05 in each). GLUT4 was abundantly expressed in soleus muscle (Fig 6D. main effect of muscle type, P <0.05), while group difference was not evident. HK2 expression is significantly higher in soleus muscle than in gastrocnemius muscle (Fig 6E, P <0.05). In addition, the RAP group showed significantly lower HK2 expression in both soleus and gastrocnemius muscle than in CON and PF groups (P <0.05 in each).

## Discussion

In this study, we investigated the effects of chronic rapamycin administration and rapamycin-induced reduction of spontaneous energy intake on skeletal muscle mass, function, and regulatory factors (signaling involves protein synthesis, degradation, and glucose metabolism) activity and/or expression in young rats. The main findings of this study are as follows: 1) Rapamycin administration showed significantly lower skeletal muscle size with decreased mTORC1-p70S6K signaling. 2) Energy restriction equal to caloric intake reduced by rapamycin administration (PF group) improved muscle contractile capacity, whereas rapamycin administration had a mostly harmless or slightly negative impact. 3) Chronic rapamycin administration induced systemic glucose intolerance, despite reduced adiposity. 4) Rapamycin

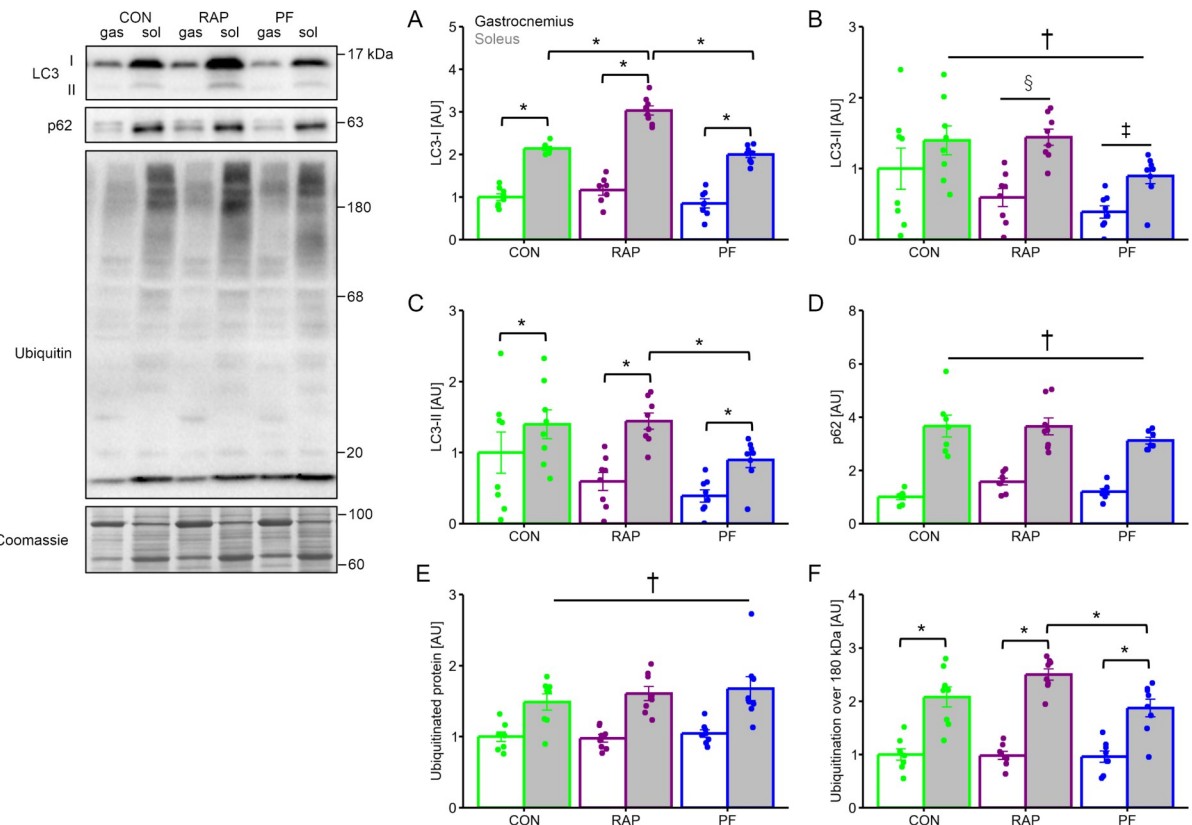

**Fig 5. The influence of 4-weeks of rapamycin administration on the muscle protein synthesis and molecules involved in protein degradation.** A. LC3-I protein expression. B. LC3-II protein expression. C. LC3-I/II expression ratio. D. p62 protein expression. E. Ubiquitinated protein expression. F. Expression of ubiquitinated proteins with a molecular weight of more than 180 kDa. n = 8 in each group. Values are expressed as MEANS ± SE with individual value. †: $P < 0.05$ between gastrocnemius vs. soleus (main effect of muscle type by 2-way ANOVA), ‡: $P < 0.05$ vs. CON group (main effect of group by 2-way ANOVA, comparison between groups). §: $P < 0.05$ vs. PF (main effect of group by 2-way ANOVA, comparison between groups). Bars with asterisks indicate significant differences ($P < 0.05$) between two linked factors.

treatment decreased Rictor expression in glycolytic and oxidative muscles and decreased mTORC2-Akt signaling and HK2 expression.

Here, we observed that rapamycin administration at doses commonly used to inhibit mTOR ($\sim 2$ mg/kg body weight) [12] resulted in lowering body growth response concomitant with reduced spontaneous feeding as previously reported [17, 18]. Fat mass in the rapamycin-administered group was not statistically different to that of the PF group, which had equivalent caloric intake, suggesting that rapamycin administration-induced fat mass reduction is largely due to reduced energy intake. Body weight-normalized skeletal muscle mass was muscle type-dependently affected by either rapamycin administration or calorie restriction. The mass of the glycolytic gastrocnemius muscle was slightly but significantly greater in the caloric-restricted PF group when compared with that in the CON group, whereas the RAP group showed a smaller size than that of the PF group. Furthermore, histological analysis showed that rapamycin administration demonstrated significantly smaller myofiber size specifically in MyHC type 1 and type 2x/b fiber independently of the energy intake. Considering with the age of rats used in this study was still growing, rapamycin would inhibit skeletal muscle growth as previously observed and it may be dependent on muscle fiber type. Meanwhile, given the rats in the PF group did not show skeletal muscle growth inhibition, rapamycin would inhibit

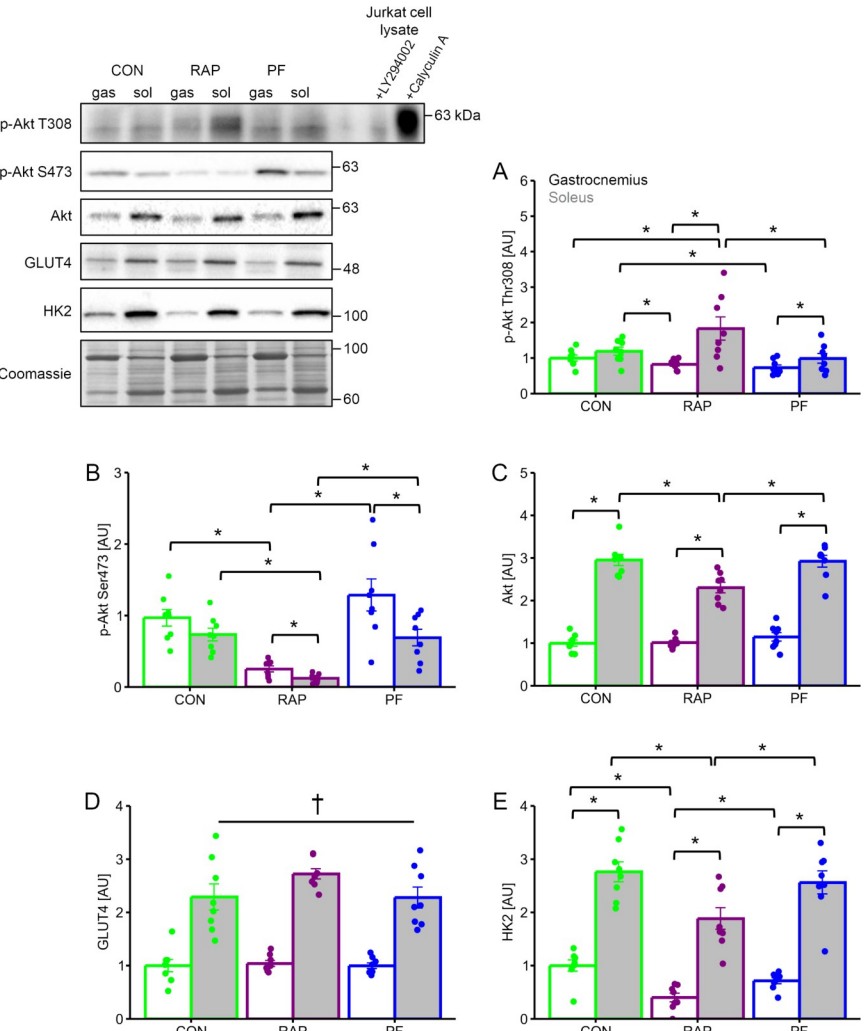

**Fig 6. The influence of 4-weeks of rapamycin administration on the glucose metabolism regulators.** A. Thr308 phosphorylated Akt protein expression. B. Ser473 phosphorylated Akt protein expression. C. Akt protein expression. D. GLUT4 protein expression. E. HK2 protein expression. n = 8 in each group. Values are expressed as MEANS ± SE with individual value. †: P <0.05, gastrocnemius vs. soleus (main effect of muscle type by 2-way ANOVA). Bars with asterisks indicate significant differences (P <0.05) between two linked factors.

skeletal muscle growth independently of the caloric intake. In contrast, although we could not perform histological analysis on soleus muscle here, the mass of the oxidative soleus muscle and type I fiber size in gastrocnemius were unaffected by rapamycin administration suggesting that the muscle growth-inhibiting effect of rapamycin might be muscle type-dependent as well as muscle fiber type. Interestingly, the most recent mice study showed that rapamycin administration and calorie restriction in aging conditions improves type IIa and IIx fiber CSA while it decreased IIb fiber CSA in fast fiber type dominant forelimb muscle. On the contrary, soleus muscle fast fiber (type 2b/x) CSA did not affect by rapamycin [22]. Thus, the effects of rapamycin on skeletal muscle may be fiber type and/or muscle type dependent and have a specific positive effect on aging muscle.

The rats subjected in this study were in growth and growth of myofiber size seems to the main contributor to muscle tissue size gain [23]. The myofiber size is generally considered to be regulated by the balance between protein synthesis and breakdown, and rapamycin is

known to inhibit resting muscle protein synthesis [14, 24, 25] suggesting that present skeletal muscle growth inhibition of rapamycin would be mediated by inhibition of muscle protein synthesis. However, although mTORC1 activity is suppressed while the protein synthesis and protein expression related to autophagy and ubiquitin-proteasomal system were almost unaltered in the RAP group in this study. This may be due to the sampling timing that in this study, samples were taken 48 hours after the last administration of rapamycin (half-life of rapamycin in rats is ∼30 hours [26]). Although we only observed the increase in LC3-I expression, we previously observed that the acute administration of rapamycin increased basal LC3-I and II expressions [24]. Given that the mTOR mechanistically suppresses autophagy [27], decreased autophagy suppression by rapamycin might potentially contribute to skeletal muscle growth inhibition. However, despite rapamycin inhibiting muscle protein synthesis, there have been no reports of rapamycin causing a decrease in adult muscle mass. This suggests that there may be a mechanism other than the effect of rapamycin on muscle protein metabolism involved in the inhibitory effect of rapamycin on muscle growth.

Interestingly, the calorie-restricted PF group showed significantly improved plantar flexor function compared to the CON group, whereas rapamycin administration failed to elicit this effect. It has previously been reported that caloric restriction is able to improve muscle function in young rats and our present observation supported for these reports [28, 29]. However, despite rapamycin administration reducing energy intake and adiposity, muscle contractile capacity failed to improve, suggesting that rapamycin treatment in young adults may have a detrimental effect on muscle contractile function as well as muscle mass growth.

The mechanisms underlying the muscle type-dependent effects of rapamycin on muscle mass and decrease in contractile force are currently unclear. However, in the soleus muscle, we observed increases in myosin heavy chain expression and the expression of ubiquitinated proteins, especially in the high-molecular-weight region (>180 kDa). We observed greater suppression of mTORC1-p70S6K signaling to rapamycin in the oxidative soleus muscle than in the glycolytic gastrocnemius muscle, and rapamycin administration may cause aberrant myosin heavy chain (Myh7) expression in soleus. Interestingly, the rapamycin-inducing Myh7 expression has also been observed in cardiac myocytes *in vitro* [30]. Thus, aberrant myosin expression might relate to the protein quality and function in slow muscle *in vivo*. Therefore, further studies are required to elucidate the mechanisms underlying the changes in contractile function following rapamycin administration and calorie restriction.

Chronic rapamycin administration (∼2mg / g bodyweight) has been reported to cause insulin resistance within 2 weeks in exchange for lifespan expansion and reduced adiposity [12, 17, 31]. We also observed that 4 weeks of rapamycin administration triggered glucose intolerance in young adult rats. In addition, systemic insulin concentrations were lower in the RAP group, suggesting that reduced insulin-secreting capacity may be associated with rapamycin-induced insulin resistance as previously reported [32, 33]. However, we did not observe any effect of rapamycin administration on fasting blood glucose levels, despite the reduced resting insulin secretion, suggesting that some redundancy complements the reduction in insulin secretion, at least in the fasted basal state. Impaired glucose uptake in some insulin-responsive tissues is considered a candidate mechanism for rapamycin-induced insulin resistance. Lamming et al. suggested diminished glucose metabolism via reduced Rictor expression in the liver as a mechanism underlying rapamycin-induced glucose intolerance [12]. Although Lamming et al. did not observe skeletal muscle glucose intolerance after 2 weeks of rapamycin administration in mice (2 mg /g per day for 2 weeks), muscle-specific deletion of Rictor in mice led to glucose intolerance and a reduction in HK2 expression in adults (3–5 months of age) [14, 19]. We could not assess glucose uptake in skeletal muscle in this study; however, HK2 expression was significantly reduced by the rapamycin administration in both the

oxidative soleus muscle and glycolytic gastrocnemius muscle. Therefore, the current results suggest that aberrant glucose metabolism might occur in skeletal muscle after 4 weeks or more of rapamycin administration in adulthood.

In summary, rapamycin administration has the advantage of reducing adiposity concomitant with reduced energy intake while it significantly inhibits skeletal muscle growth with myofiber and muscle type-dependent response. We also showed that administration of rapamycin in young adulthood can diminish skeletal muscle function and metabolism with muscle-type-dependent pharmacological responsivity. Emerging evidence indicated that rapamycin has a beneficial additive effect to calorie restriction on senescent skeletal muscle. Considering the current observation, the influence of rapamycin on skeletal muscle might largely be distinct in different life stages. Understanding the long-term effects of rapamycin administration on peripheral tissues, including skeletal muscle, may provide strategies for developing calorie restriction mimetics that maintain general health.

## Material and methods

### Animal and experimental procedure

The experimental procedures used in this study were approved by the Ethics Committee for Animal Experiments at Nagoya Institute of Technology, conducted in accordance with the Institutional Regulations for Animal Experiments and Fundamental Guidelines for Proper Conduct of Animal Experiments and Related Activities in Academic Research Institutions under the jurisdiction of the Ministry of Education, Culture, Sports, Science and Technology of Japan, and followed the ARRIVE guidelines. Male Sprague-Dawley rats aged 10-week were purchased from Japan SLC. The rats were acclimated for a week in an environment maintained at 22°C with an alternating 12h light-dark cycle. A standard rodent diet CE-2 (CLEA Japan, Inc., Tokyo, Japan) and water were given ad libitum. After the acclimation, the rats were randomly assigned control (CON) group, rapamycin (RAP) group, and pair-fed (PF) group (n = 8 in each group). RAP group was administrated rapamycin 1.5 mg/kg body weight (0.25 mg/ml in 0.5% DMSO/PBS) intraperitoneally, 3 times a week for 4 weeks (28 days). The dose and schedule of rapamycin administration were determined based on our previous studies examining the effects of rapamycin on skeletal muscle adaptation to muscle contraction [24], and the dosage is about that commonly used for mTOR inhibition [12, 34]. CON group and PF group were administrated vehicles (DMSO/PBS) in an identical volume and timing. PF group was fed to similar food intake to the RAP group (pair-feeding). In detail, CON and RAP groups began the experiment a week before that starting the intervention of the PF group. PF group fed the average amount of diet that was monitored before a week in the RAP group (18.45 g) consistently during the experimental period. Therefore, the RAP and PF groups were restricting their calorie intake by ∼22% during the experiment. Intraperitoneal glucose tolerance test and muscle force measurement was performed on the day 25$^{th}$. After the 28 days of intervention, 2 days of washout was done to minimize the acute effect of rapamycin administration. On day 30$^{th}$, the rats were 12 h fasted and sacrificed by blood removal from the abdominal aorta. The medial head of the gastrocnemius muscle and the soleus muscle were dissected and rapidly frozen in liquid nitrogen. The lateral head of gastrocnemius were covered with O.C.T compound and frozen in cooled isopentane. Collected blood samples were centrifuged for plasma and serum. The samples were kept at -80°C until analysis.

### Intraperitoneal glucose tolerance test (IPGTT)

IPGTT was performed as previously described [35]. Briefly, after an overnight fast, the rats were anesthetized with inhaled isoflurane, and baseline blood glucose levels were measured. A

glucose (2 g/kg body weight) was injected in abdominal and release from anesthesia. After the glucose administration, the blood glucose levels were monitored until 2 h after under awakefulness. Blood glucose levels were measured with Glucose Pilot (Iwai chemicals company, Tokyo, Japan).

## Muscle force measurement

The rats were immobilized on a stand under isoflurane inhalation anesthesia, and the right foot was attached to a footplate to which a force transducer was connected. (cat#T.K.K.5813, Takei scientific instruments, Niigata, Japan) with the tibia-foot angle to be 90˚. Surface electrodes were attached to the medial side of lower leg and Achilles tendon so that electrical stimulation covered the triceps surae muscle, and triceps surae muscle contraction was evoked via percutaneous electrical stimulation. Twitch force (single 1 ms square pulse) and tetanus force (60 times of 1 ms pulse with 10 ms interval) were measured with optimized voltage ($\sim$30V) to be maximal force production. Measured values were used to evaluate the maximal contractile ability.

## *in vivo* SUnSET

*in vivo* SUnSET was performed as previously reported with slight modification [36]. Briefly, puromycin dissolved in PBS (0.02 M) was intraperitoneally injected into the rats (0.04 $\mu$mol g$^{-1}$ body weight) and killed just after 15 min. The puromycilated peptide was detected with western blot as described below.

## Insulin concentration measurement

Serum insulin concentration was measured using LBIS Rat Insulin ELISA Kit U-E type (FUJI-FILM Wako Shibayagi) according to the manufacturer's instruction.

## Western blot

Powdered frozen tissue was homogenized with radioimmunoprecipitation assay (RIPA) buffer supplemented with protease inhibitor and phosphatase inhibitor. The crude homogenate was centrifuged at 2,000 $\times$ g for puromycilated protein detection or at 10,000 $\times$ g for signaling molecules detection. The protein concentration of the resulting supernatant was measured by Protein Assay Rapid Kit Wako II (FUJIFILM WAKO, Osaka, Japan) and boiled with Laemmle sample buffer for 5min at 95˚C. The sample (10 $\mu$g protein for total protein detection or 30 $\mu$g protein for phosphorylated protein detection) was resolved with electrophoresis and then transferred to the ClearTrans$\circledR$ SP PVDF Membrane (FUJIFILM WAKO). The transferred membrane was blocked with 5% skim milk in TBST for 30 min. After the sequential wash with TBS-tween (TBST), the membrane was incubated with primary antibody overnight on a rocker at 4˚C. The next day, the membrane was incubated with an HRP-conjugated secondary antibody for 90 min after a sequential wash with TBST. Chemiluminescence was detected on the ChemiDoc XRS+ system (Bio-Rad, Hercules, CA, US). After the detection, the antibody that bound on the membrane was stripped off with stripping solution (FUJIFILM WAKO) and stained with Coomassie brilliant blue (CBB). The protein band intensity was measured with Image Lab$^{TM}$ version 6.0.1 (Bio-Rad) and normalized by total protein band (CBB) intensity. The primary antibodies used here are listed below. mTOR (1:1000, cat#2983, Cell signaling technology, Danvers, MA), FKBP12 (1:500, cat#sc-133067, Santa Cruz Biotechnology, Dallas, TX), Raptor (1:1000, cat#2280, CST), Rictor (1:1000, cat#2114, CST), p-p70S6K T389 (1:1000, cat#9205, CST), p70S6K (1:1000, cat#2708, CST), p-4EBP1 T37/46 (1:1000, cat#9459, CST), 4EBP1 (1:1000, cat#9644, CST), p-rpS6 S240/244 (1:1000, cat#2215, CST), rpS6 (1:1000,

cat#2217, CST), LC3 (1:1000, cat#2775, CST), p62 (1:1000, cat#PM045, MEDICAL & BIO-LOGICAL LABORATORIES, Tokyo, Japan), Ubiquitin (1:1000, cat#3933, CST), p-Akt S473 (1:1000, cat#9271, CST), p-Akt T308 (1:1000, cat#13038, CST), Akt (1:1000, cat#9272, CST), GLUT4 (1:1000, cat#07–1404, Millipore, Darmstadt, Germany), HK2 (1:1000, cat#2876, CST). For the original unprocessed blot image, see S1 Fig.

### Myosin heavy chain typing and content analysis with SDS-PAGE

Myosin heavy chain typing was performed as previously described [37]. Briefly, 10 ug of the protein samples prepared for the western blot were resolved with SDS-PAGE. After electrophoresis, the protein within gels was visualized with CBB staining. The protein band of the myosin heavy chain was scanned with ChemiDoc XRS+ system (Biorad) and analyzed using Image Lab$^{TM}$ version 6.0.1 (Bio-Rad). The band intensity of each myosin heavy chain type was quantified, and the composition of myosin heavy chain types was determined from the composition as a percentage of the intensity of each myosin heavy chain type relative to the intensity of all myosin bands. For the original unprocessed gel image, see S1 Fig.

### Immunohistochemistry

The 6 μm cross-section of the lateral head of gastrocnemius was prepared with a cryostat. The sections were attached with a glass slide and air dried for 20min. The sections were blocked with 5% normal goat serum, and 0.3% Triton-X 100 containing PBS for 1 h at room temperature. After the sequential wash with PBS, the sections were incubated with primary antibody against Myh2 (1:200, clone: SC71, Developmental Studies Hybridoma Bank (DSHB), Iowa city, IA), Myh7 (1:50, clone: BA-D5, DSHB), and laminin diluted in 5% BSA containing PBS for overnight at 4˚C. After the reaction with the primary antibody, sections were incubated with the appropriate secondary antibody that conjugated with the Alexa Fluor® 350 (Myh7), 488 (Myh2), and 555 (Laminin) for 90 min at room temperature. After the sequential wash with PBS, the slides were mounted with VECTASHIELD Antifade Mounting Medium.

The sections were observed with ×10 objectives equipped with epi-fluorescent microscope BZ-X 700 (Keyence, Osaka, Japan), random 3 points of surface and deep part regions were respectively photographed (each image contained ∼300 fibers, total ∼2000 fibers). Image processing was performed with Fiji (ImageJ ver. 1.53,). The images that visualized laminin were used for myofiber cross-sectional area (CSA) measurement with particle analysis plugin. The images that visualized Myh2 and Myh7 were auto-thresholded and analyzed for myosin heavy chain type determination. The images data analysis was performed with blind manner.

### Statistical analysis

Two-way ANOVA (group × time or muscle type) was used to evaluate the changes in body weight, food consumption, IPGTT, and protein expression. Post hoc comparison was performed with paired t-test with Benjamini-Hochberg false discovery rate when interaction was observed. One-way ANOVA was used to compare the tissue weight, MHC type composition, and myosin content, and post hoc comparison was performed with paired t-test with Benjamini-Hochberg false discovery rate when the main effect was observed. Statistical significance was set at P <0.05 in this study.

## Supporting information

**S1 Fig. Original uncropped images of gel and western blot used for Figs 2–6.** All images were photographed using ChemiDoc XRS+ system (Bio-Rad, Hercules, CA, US). The region

shown in the figures were indicated with dashed line.
(PDF)

## Acknowledgments

Dr. Riki Ogasawara passed away before the submission of the final version of this manuscript. Corresponding author Satoru Ato accepts responsibility for the integrity and validity of the data collected and analyzed.

## Author Contributions

**Conceptualization:** Riki Ogasawara.

**Data curation:** Satoru Ato, Chieri Oya.

**Formal analysis:** Satoru Ato, Chieri Oya, Riki Ogasawara.

**Investigation:** Satoru Ato.

**Methodology:** Satoru Ato.

**Supervision:** Satoru Ato.

**Validation:** Satoru Ato, Chieri Oya.

**Visualization:** Satoru Ato.

**Writing – original draft:** Satoru Ato, Riki Ogasawara.

**Writing – review & editing:** Satoru Ato, Riki Ogasawara.

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
