## [Decision Letter · Decision Letter 0]

9 Aug 2024

PONE-D-24-27587Effects of rapamycin administration on skeletal muscle function and metabolism in young adult rats.PLOS ONE

Dear Dr. Ato,

Thank you for submitting your manuscript to PLOS ONE. After careful consideration, we feel that it has merit but does not fully meet PLOS ONE’s publication criteria as it currently stands. Therefore, we invite you to submit a revised version of the manuscript that addresses the points raised during the review process.

We look forward to receiving your revised manuscript.

Kind regards,

Kai-Hei Tse

Academic Editor

PLOS ONE

Journal Requirements:

2. We note that your Data Availability Statement is currently as follows: All relevant data are within the manuscript and its Supporting Information files

Reviewers' comments:

Reviewer's Responses to Questions

**Comments to the Author**

1. Is the manuscript technically sound, and do the data support the conclusions?

Reviewer #1: Yes

Reviewer #2: Yes

Reviewer #3: Partly

Reviewer #4: Partly

2. Has the statistical analysis been performed appropriately and rigorously? 

Reviewer #1: No

Reviewer #2: Yes

Reviewer #3: Yes

Reviewer #4: Yes

3. Have the authors made all data underlying the findings in their manuscript fully available?

Reviewer #1: Yes

Reviewer #2: Yes

Reviewer #3: Yes

Reviewer #4: Yes

4. Is the manuscript presented in an intelligible fashion and written in standard English?

Reviewer #1: Yes

Reviewer #2: Yes

Reviewer #3: Yes

Reviewer #4: No

5. Review Comments to the Author

Reviewer #1: This is an interesting study of the effects of rapamycin on skeletal muscle. All experiments are carefully conducted and contain enough numbers of rats. But, please address some minor concerns.

Abstract

RAPA notation is used in the abstract, but RAP notation is used in the results, so it is recommended to unify them.

Line 21

Please add the citation into "Skeletal muscle loss occurs under several conditions, including aging, cancer cachexia, and type 2 diabetes".

Line 54

What is PF, and CON? Please correct the abbreviation because Methods is behind results section.

Are Figure 2H-I quantified CBB above?

Figure 2G

Please provide a description of which band is which myosin next to the photo.

Figure 5E.

If P=0.1, isn't it significantly?

Figure 3.

Coomassie's photo is small, is the photo being shrunk? Or is it showing the result of dyeing the local membrane?

Line 373

Riki Ogasawara is not the corresponding author in this manuscript.

Reviewer #2: This is an important paper that presents new information from a well-designed study. The study aimed to compare the effects of chronic rapamycin treatment and calorie restriction on skeletal muscle. I have a few comments as follows:

1. Title (and throughout the manuscript): The word “metabolism” seems to be too broad. This is because, as mentioned in the results section, the authors only measured some of the “glucose metabolism regulators”. I suggest measuring some key indicators of glycolytic and oxidative markers if samples are still available.

2. Line 4: Is the term “mimetics” appropriate given that the impact of CR on mTOR signalling is clearly different from that of RAPA?

3. Line 13: Does the term “similar” correctly reflect the results? It does not appear to be consistent with the statements in the later discussion (Line 212-213).

4. Line 27: Please include an important reference from your own group on calorie restriction and skeletal muscle mitochondria (PMID: 27539498).

5. Line 203: As the RAPA group is not obese, this discussion may not be necessary.

6. Line 269: Please indicate what percentage of calories were restricted in this study.

7. Line 373: This may depend on the policy of the journal, but it would have to be someone else, as RO has already passed away.

8. Figures: Since the dots indicating statistical results and individual data are sometimes confusing, it would be a good idea to use color as this is an online journal.

Reviewer #3: Rationale for Significance Rating for Authors

The authors described the effects of chronic rapamycin administration compared to caloric restrictions on the muscle growth of early adult rats by analysing the longitudinal change of muscle mass and protein expressions associated with mTORC composition to suggest that rapamycin administration could reduce muscle mass and growth similar to reduced calorie intake by reducing mTORC signalling.

Reviewer Commentary

This manuscript by Ato et al. shows a similar reduction of muscle mass between rapamycin administration and caloric restrictions through alterations of mTOR inhibitor composition. The results show evident decrease in the weight of RAPA rats with corresponding reduction in food intake and serum insulin concentrations, suggesting reduced energy intake as the reasoning for this decrease. Muscle fiber quantification further show decreased fiber cross-sectional area and tetanic force from RAPA rats.

Gastrocnemius and soleus muscles were then analysed for protein expressions of mTORC components to show possible inhibition of mTORC-driven regulations that could affect skeletal muscle growth. While the gastrocnemius muscle did not show much effect among diet groups, the soleus muscle showed a significant reduction in the RAP rat model in mTORC2 components, Rictor, and phosphorylated mTORC1 residue, p70S6K Thr389. Furthermore, autophagy marker, LC3-I, was shown in reduced expression in RAP rat. The authors conclude that rapamycin administration reduces muscle-type-specific growth with varying age-dependent effects. With the majority of the manuscript supported by downstream events of the hypothesis, it may still require mechanistic insight to link up the cause (rapamycin or calorie restriction) and effect (weight loss, reduced muscle loss, and reduced protein expression).

Major Concerns

1. One of their conclusions is that the effect of rapamycin is variable among ages. The use of 10-week rats indicates their knowledge of their pre-adulthood growth and the factors that may affect the experiment, especially in skeletal muscle growth. Correspondingly, their results of weight changes (Fig. 1A) indicate that by 22-25 days post drug administration, the weights of CON and RAP rat decreases, showing an unprecedented trend in a young rat. Please explain the decision on the age choice of rat compared to a more stable age such as 5 or 6 months.

2. Many of their significant differences were seen in the soleus muscle, not the gastrocnemius muscle, yet the immunohistochemistry of fiber types and their CSA were from the gastrocnemius muscle. It would be beneficial for there to be similar data for the soleus muscle to connect the significant changes seen in the Western blots with changes in fiber area.

3. The use of Western blot to quantify proteins to suggest changes in the activity of mTOR is weak. They stated the relationship among autophagy of skeletal muscle cells, LC3-I and II expression, and rapamycin. This is strong supporting data if performed and will strengthen their hypothesis of the possible pathway resulting in the muscle reduction. Currently, they do not have any mechanistic nor proposed pathway of rapamycin’s effect on muscle growth.

4. According to Lines 203-206, they acknowledged the unprecedented obesity of their control group by suggesting that the calorie deficit “returned” the rat to normal weight, resulting in improved muscle function. If this is the case, then no conclusion nor comparisons could be drawn from any of their figures as the control group is no longer valid.

5. Their observations of the reduced mTORC components in RAP rats suggest alterations that may inhibit mTORC activity. What are the downstream effects of these changes? As mTOR is upstream of p70S6K and Rictor, why is mTOR unaffected, yet p70S6K and Rictor are significantly reduced in RAP rats? There are many holes that need to be filled to relate these events back to mTOR, and ultimately, muscle growth.

6. The curve for the PF group for Figure 1B is missing. Please add it back in.

Reviewer #4: Satoru et al. aimed to investigate effects of chronic rapamycin administration on skeletal muscle function and metabolism in young adult rats. The background for performing this study is associated to reported effects of rapamycin resembling those observed under calorie restriction. The authors found that rapamycin administration showed reduced skeletal muscle size with decreased mTORC1-p70S6K signaling, slightly negative impact on muscle contractile capacity and decreased Rictor expression in glycolytic and oxidative muscles and decreased mTORC2-Akt signalling and HK2 expression.

The paper presents some novel and interesting data by using a rat model, which might represent a valuable addition to the field. The experiments appear to be well-performed. Unfortunately, current manuscript gives a slightly unpolished overall impression, making it somewhat difficult to comprehend, but also to really assess the value of the presented data. My main advice would be to carefully go through all technical aspects of the manuscript, proof-read and make sure it is understandable. Here are my major concerns that need to be addressed by the authors for the paper to be considered for publication:

1. There are inconsistencies in the abbreviations used in the introduction part vs. the rest of the manuscript (RAPA vs. RAP; CR used in the abstract is not mentioned again in the text). Given the chronological order of the current text, it would be helpful for the reader if list of abbreviations was provided, or if the abbreviations would be explained first time they are used in the text.

2. The authors should also provide more details on the feeding protocol under materials and methods (the composition of the food in the different groups).

3. How many rats were included in the present study? What is the number of replicates that the statistical analysis is based on? Please provide a more detailed information under materials and methods, and this should also be specified in each figure legend.

4. In figure 1B, results for PF group are not presented. The authors need to clarify why these data are not presented.

A general comment to figure legends is that they need to be more detailed than they are in the submitted manuscript. A figure legend should be self-explanatory, providing sufficient amount of information for figure to be able to stand independently and yet be understood.

5. Figure 2: please provide description of the presented immunoblot. I assume this is a representative blot, but a detailed description should be included in the figure legend.

6. In several figures, the authors present “ME”, which is abbreviation for “Main effect”. Please specify the meaning of this term in each case. Also, in figure 5B, the authors use “ME muscle type and ME group”. Please specify the meaning of these terms.

7. In several figure legends, the authors write “different character indicates statistically significant difference”. I strongly encourage the authors to avoid this kind of unspecific terms and in stead specify each specific character that is used to label statistical differences.

6. PLOS authors have the option to publish the peer review history of their article (what does this mean?). If published, this will include your full peer review and any attached files.

Reviewer #1: **Yes**

Reviewer #2: No

Reviewer #3: **Yes**

Reviewer #4: No

---

## [Author Response · Author response to Decision Letter 0]

23 Sep 2024

Reviewer #1: This is an interesting study of the effects of rapamycin on skeletal muscle. All experiments are carefully conducted and contain enough numbers of rats. But, please address some minor concerns.

We thank you for reading through the paper in detail and providing insightful and thought-provoking comments.

Abstract

RAPA notation is used in the abstract, but RAP notation is used in the results, so it is recommended to unify them.

>Thank you very much for your helpful indication. We again carefully read through the manuscript and unified the abbreviation that appeared in the manuscript.

Line 21

Please add the citation into "Skeletal muscle loss occurs under several conditions, including aging, cancer cachexia, and type 2 diabetes".

>I have followed the reviewer's suggestion and added the citation to the line 22.

Line 54

What is PF, and CON? Please correct the abbreviation because Methods is behind results section.

＞Thanks for pointing out these helpful corrections. Abbreviations have been corrected throughout the manuscript.

Are Figure 2H-I quantified CBB above?

>We appreciate your important remarks. As you are aware, fig.2H,I are quantitative data for the bands shown in the upper part of Fig.2G. We did not include this information in the current manuscript, so we have included it in the material & method section and figure legend.

Figure 2G

Please provide a description of which band is which myosin next to the photo.

>Thank you for your HELPFUL comments. I have followed your suggestion and added a caption next to the band images indicating which myosin type each band is.

Figure 5E.

If P=0.1, isn't it significantly?

> We appreciate your remarks on this very important point. The p=0.1 is shown in the figure because it is a trend of interest as a factor that could explain the decrease in muscle strength with rapamycin administration, although it is not significant by the criteria (p <0.05) we set. On the other hand, we excluded this expression from the revised figure because it could be confusing.

Figure 3.

Coomassie's photo is small, is the photo being shrunk? Or is it showing the result of dyeing the local membrane?

>The part of the results for the signalling proteins is too large to describe all the CBB-stained membranes, so a small portion is shown in the figure. Therefore, the molecular weights of the indicated parts have been added.

Line 373

Riki Ogasawara is not the corresponding author in this manuscript.

> Much appreciated. This is a typographical error and has been corrected.

Reviewer #2: This is an important paper that presents new information from a well-designed study. The study aimed to compare the effects of chronic rapamycin treatment and calorie restriction on skeletal muscle. I have a few comments as follows:

We would like to thank you for taking the time to read our paper in detail and for providing us with useful suggestions for improvement.

1. Title (and throughout the manuscript): The word “metabolism” seems to be too broad. This is because, as mentioned in the results section, the authors only measured some of the “glucose metabolism regulators”. I suggest measuring some key indicators of glycolytic and oxidative markers if samples are still available.

> Thank you for your important point. As you pointed out, the expression “metabolism” is abstract, and there are many aspects that are not covered by the results of this study, so we have revised the title as follows.

“Rapamycin administration causes a decrease in muscle contractile function and systemic glucose intolerance concomitant with a decrease in skeletal muscle Rictor, the mTORC2 component expression independent of energy intake in young rats”

Unfortunately, we do not have any soleus muscle samples available for further analysis, but the data we already have on mitochondrial respiratory chain complex protein expression show that ~22% caloric restriction had no effect on OXPHOS expression was observed.

2. Line 4: Is the term “mimetics” appropriate given that the impact of CR on mTOR signalling is clearly different from that of RAPA?

>Thank you for your question regarding this important point. As you pointed out, based on the present results, the effects of CR and rapamycin are clearly different. However, as discussed in Unnikrishnan et al. J Gerontol A Biol Sci Med Sci 2020, there has been a general focus on the possibility that rapamycin mimics CR, and I mention this in the introduction.

3. Line 13: Does the term “similar” correctly reflect the results? It does not appear to be consistent with the statements in the later discussion (Line 212-213).

>Thank you for pointing this out. The description in the abstract was incorrect, so we have made the necessary corrections.

4. Line 27: Please include an important reference from your own group on calorie restriction and skeletal muscle mitochondria (PMID: 27539498).

> We forgot to cite our own important research. We thank you for your kind support.

5. Line 203: As the RAPA group is not obese, this discussion may not be necessary.

>Thank you for pointing this out. Reviewer#3 also pointed this out, and when we looked into previous research, we found that there were multiple reports of studies observing that muscle function improves with calorie restriction, and we considered that we had also observed a similar effect. Therefore, we have excluded the statement you pointed out.

6. Line 269: Please indicate what percentage of calories were restricted in this study.

>We added the detailed explanation in revised manuscript line 272-273.

7. Line 373: This may depend on the policy of the journal, but it would have to be someone else, as RO has already passed away.

>Thank you for pointing this out.

We checked with the journal's office and were told that we can be listed as a deceased author after filling in the necessary information in the acknowledgements. Therefore, we will add Riki Ogasawara as an author in the acknowledgements, stating that he is deceased.

8. Figures: Since the dots indicating statistical results and individual data are sometimes confusing, it would be a good idea to use color as this is an online journal.

> Thank you for your very constructive suggestions. I thought using color was a very good idea. I have followed your suggestion and modified the data to a color figure. Hopefully this change will make the data easier to understand.

Reviewer #3: Rationale for Significance Rating for Authors

The authors described the effects of chronic rapamycin administration compared to caloric restrictions on the muscle growth of early adult rats by analysing the longitudinal change of muscle mass and protein expressions associated with mTORC composition to suggest that rapamycin administration could reduce muscle mass and growth similar to reduced calorie intake by reducing mTORC signalling.

Reviewer Commentary

This manuscript by Ato et al. shows a similar reduction of muscle mass between rapamycin administration and caloric restrictions through alterations of mTOR inhibitor composition. The results show evident decrease in the weight of RAPA rats with corresponding reduction in food intake and serum insulin concentrations, suggesting reduced energy intake as the reasoning for this decrease. Muscle fiber quantification further show decreased fiber cross-sectional area and tetanic force from RAPA rats.

Gastrocnemius and soleus muscles were then analysed for protein expressions of mTORC components to show possible inhibition of mTORC-driven regulations that could affect skeletal muscle growth. While the gastrocnemius muscle did not show much effect among diet groups, the soleus muscle showed a significant reduction in the RAP rat model in mTORC2 components, Rictor, and phosphorylated mTORC1 residue, p70S6K Thr389. Furthermore, autophagy marker, LC3-I, was shown in reduced expression in RAP rat. The authors conclude that rapamycin administration reduces muscle-type-specific growth with varying age-dependent effects. With the majority of the manuscript supported by downstream events of the hypothesis, it may still require mechanistic insight to link up the cause (rapamycin or calorie restriction) and effect (weight loss, reduced muscle loss, and reduced protein expression).

We would like to express our gratitude for taking the time to provide insightful comments and suggestions on our paper.

Major Concerns

1. One of their conclusions is that the effect of rapamycin is variable among ages. The use of 10-week rats indicates their knowledge of their pre-adulthood growth and the factors that may affect the experiment, especially in skeletal muscle growth. Correspondingly, their results of weight changes (Fig. 1A) indicate that by 22-25 days post drug administration, the weights of CON and RAP rat decreases, showing an unprecedented trend in a young rat. Please explain the decision on the age choice of rat compared to a more stable age such as 5 or 6 months.

>Thank you for pointing out this important point. Although we did not make a direct comparison in this study, and therefore cannot say for certain whether there is a difference between the effects in more mature individuals (5-6 months of age), it is likely that the growth-inhibitory effects are reduced in more mature individuals. On the other hand, since a previous study on old individuals reported that rapamycin treatment prolongs life span while inducing systemic glucose intolerance, such effects may occur in mature individuals as well. In the present study, we used the same week-old rats as those used in our previous study to clarify the effects of rapamycin administration itself on resting skeletal muscle, while we have been studying the role of mTORC1 in muscle hypertrophy induced by muscle contraction using rapamycin. We used the same week-old rats as those used in our previous study.

2. Many of their significant differences were seen in the soleus muscle, not the gastrocnemius muscle, yet the immunohistochemistry of fiber types and their CSA were from the gastrocnemius muscle. It would be beneficial for there to be similar data for the soleus muscle to connect the significant changes seen in the Western blots with changes in fiber area.

>Thank you for your important remarks. Unfortunately, we are unable to provide the data because we do not have any soleus left available for histochemistry. This point has been noted as a limitation in the discussion line 181-182. However, since there is no effect of rapamycin on the size of the soleus muscle or the size of the Type I fibers of the gastrocnemius muscle, the effect of rapamycin on the size of slow-twitch muscle may not be significant. 

3. The use of Western blot to quantify proteins to suggest changes in the activity of mTOR is weak. They stated the relationship among autophagy of skeletal muscle cells, LC3-I and II expression, and rapamycin. This is strong supporting data if performed and will strengthen their hypothesis of the possible pathway resulting in the muscle reduction. Currently, they do not have any mechanistic nor proposed pathway of rapamycin’s effect on muscle growth.

> We thank you for your thought-provoking remarks. Although we partially agree with the reviewer's comments, we believe that the expression of LC3 is even more indirect in terms of assessing the direct effects of mTOR activity than the phosphorylation levels of direct substrates such as p70S6K and 4E-BP1. In fact, in many studies, phosphorylation of p70S6K and 4EBP1 has been widely used as a marker of mTOR (especially mTORC1) activity, and Akt Ser473 as an indicator of mTORC2 activity (Battaglioni et al. Cell 2022), which has a certain validity. Regarding the second question, the mechanism by which rapamycin suppresses growth, although the paper mentions autophagy and protein synthesis, it is quite possible that factors other than protein metabolism are involved in growth inhibition, since muscle mass is not reduced in aged animals with rapamycin administration. Therefore, it is quite possible that factors other than protein metabolism are involved in growth inhibition. As the reviewer pointed out, this gap cannot be filled by this study alone, and it may be necessary to consider the effects on stem cells and non-muscle cells involved in muscle growth in the future study. Thus, we added the discussion in revised manuscripts line 202-205.

4. According to Lines 203-206, they acknowledged the unprecedented obesity of their control group by suggesting that the calorie deficit “returned” the rat to normal weight, resulting in improved muscle function. If this is the case, then no conclusion nor comparisons could be drawn from any of their figures as the control group is no longer valid.

> Thank you for your important remarks and inshigtful comments. Indeed, the current description was a denial of the validity of our own experimental data. Since the body composition of animals varies somewhat depending on the experimental environment, we may not be able to make a general comparison with other studies. We also overlooked an important previous study that has already observed that caloric restriction improves muscle function in young rats. While these studies used grip strength tests to assess neuro-muscular function, our current observation that tension during twitch contraction with electrical stimulation was also improved suggests that caloric restriction may have the effect of improving the contractile function of the muscle itself. Therefore, we have revised the description regarding this point.

5. Their observations of the reduced mTORC components in RAP rats suggest alterations that may inhibit mTORC activity. What are the downstream effects of these changes? As mTOR is upstream of p70S6K and Rictor, why is mTOR unaffected, yet p70S6K and Rictor are significantly reduced in RAP rats? There are many holes that need to be filled to relate these events back to mTOR, and ultimately, muscle growth.

>Thank you for your comments on this important point. The decrease in the expression of mTOR complex components by rapamycin treatment is a long-term effect, and it has been shown that a single dose of rapamycin does not alter the expression of Rictor (Ogasawara et al. J Physiol 2020). In addition, since rapamycin binds to FKBP12 and allosterically inhibits mTOR, the lack of effect on mTOR expression is not a particularly surprising phenomenon. I agree with your point that the mechanism of muscle growth inhibition by rapamycin, as mentioned above, is still open for further investigation.

6. The curve for the PF group for Figure 1B is missing. Please add it back in.

>Thank you for your suggestions. We have added data on the amount of food fed to PF groups to the revised chart.

Reviewer #4: Satoru et al. aimed to investigate effects of chronic rapamycin administration on skeletal muscle function and metabolism in young adult rats. The background for performing this study is associated to reported effects of rapamycin resembling those observed under calorie restriction. The authors found that rapamycin administration showed reduced skeletal muscle size with decreased mTORC1-p70S6K signaling, slightly negative impact on muscle contractile capacity and decreased Rictor expression in glycolytic and oxidative muscles and decreased mTORC2-Akt signalling and HK2 expression.

The paper presents some novel and interesting data by using a rat model, which might represent a valuable addition to the field. The experiments appear to be well-performed. Unfortunately, current manuscript gives a slightly unpolished overall impression, making it somewhat difficult to comprehend, but also to really assess the value of the presented data. My main advice would be to carefully go through all technical aspects of the manuscript, proof-read and make sure it is understandable. Here are my major concerns that need to be addressed by the authors for the paper to be considered for publication:

We would like to thank you for taking the time to read our paper and providing us with useful comments that will help us improve the paper as a whole.

1. There are inconsistencies in the abbreviations used in the introduction part vs. the rest of the manuscript (RAPA vs. RAP; CR used in the 

---

## [Decision Letter · Decision Letter 1]

15 Oct 2024

Rapamycin administration causes a decrease in muscle contractile function and systemic glucose intolerance concomitant with reduced skeletal muscle Rictor, the mTORC2 component, expression independent of energy intake in young rats.

PONE-D-24-27587R1

Dear Dr. Ato,

We’re pleased to inform you that your manuscript has been judged scientifically suitable for publication and will be formally accepted for publication once it meets all outstanding technical requirements.

Kind regards,

Kai-Hei Tse

Academic Editor

PLOS ONE

Additional Editor Comments (optional):

Reviewers' comments:

Reviewer's Responses to Questions

**Comments to the Author**

1. If the authors have adequately addressed your comments raised in a previous round of review and you feel that this manuscript is now acceptable for publication, you may indicate that here to bypass the “Comments to the Author” section, enter your conflict of interest statement in the “Confidential to Editor” section, and submit your "Accept" recommendation.

Reviewer #1: All comments have been addressed

Reviewer #2: All comments have been addressed

Reviewer #3: All comments have been addressed

2. Is the manuscript technically sound, and do the data support the conclusions?

Reviewer #1: Yes

Reviewer #2: (No Response)

Reviewer #3: Yes

3. Has the statistical analysis been performed appropriately and rigorously? 

Reviewer #1: I Don't Know

Reviewer #2: (No Response)

Reviewer #3: Yes

4. Have the authors made all data underlying the findings in their manuscript fully available?

Reviewer #1: Yes

Reviewer #2: (No Response)

Reviewer #3: Yes

5. Is the manuscript presented in an intelligible fashion and written in standard English?

Reviewer #1: Yes

Reviewer #2: Yes

Reviewer #3: Yes

6. Review Comments to the Author

Reviewer #1: (No Response)

Reviewer #2: (No Response)

Reviewer #3: Responses provided by the authors for comments made by reviewers were properly explained. There are no further comments from this side.

7. PLOS authors have the option to publish the peer review history of their article (what does this mean?). If published, this will include your full peer review and any attached files.

Reviewer #1: **Yes: **Keisuke Hitachi

Reviewer #2: No

Reviewer #3: No

---

## [Editor Report · Acceptance letter]

28 Oct 2024

PONE-D-24-27587R1 

PLOS ONE

Dear Dr. Ato, 

I'm pleased to inform you that your manuscript has been deemed suitable for publication in PLOS ONE. Congratulations! Your manuscript is now being handed over to our production team.

Kind regards, 

on behalf of

Dr. Kai-Hei Tse 

Academic Editor

PLOS ONE